# The Function of DNA Demethylase Gene ROS1a Null Mutant on Seed Development in Rice (*Oryza Sativa*) Using the CRISPR/CAS9 System

**DOI:** 10.3390/ijms23126357

**Published:** 2022-06-07

**Authors:** Faiza Irshad, Chao Li, Hao-Yu Wu, Yan Yan, Jian-Hong Xu

**Affiliations:** 1Zhejiang Key Laboratory of Crop Germplasm, Institute of Modern Seed Industry, Zhejiang University, Hangzhou 310058, China; faizairshad2012@yahoo.com (F.I.); 0621121@zju.edu.cn (C.L.); why1625@zju.edu.cn (H.-Y.W.); yanyan9569@126.com (Y.Y.); 2Shandong (Linyi) Institute of Modern Agriculture, Zhejiang University, Linyi 276000, China; 3Hainan Institute, Zhejiang University, Sanya 572025, China

**Keywords:** rice (*Oryza sativa*), *OsROS1a*, CRISPR/Cas9, seed storage protein, starch, RNA-Seq, alternative splicing

## Abstract

The endosperm is the main nutrient source in cereals for humans, as it is a highly specialized storage organ for starch, lipids, and proteins, and plays an essential role in seed growth and development. Active DNA demethylation regulates plant developmental processes and is ensured by cytosine methylation (5-meC) DNA glycosylase enzymes. To find out the role of *OsROS1a* in seed development, the null mutant of *OsROS1a* was generated using the CRISPR/Cas9 system. The null mutant of *OsROS1a* was stable and heritable, which affects the major agronomic traits, particularly in rice seeds. The null mutant of *OsROS1a* showed longer and narrower grains, and seeds were deformed containing an underdeveloped and less-starch-producing endosperm with slightly irregularly shaped embryos. In contrast to the transparent grains of the wild type, the grains of the null mutant of *OsROS1a* were slightly opaque and rounded starch granules, with uneven shapes, sizes, and surfaces. A total of 723 differential expression genes (DEGs) were detected in the null mutant of *OsROS1a* by RNA-Seq, of which 290 were downregulated and 433 were upregulated. The gene ontology (GO) terms with the top 20 enrichment factors were visualized for cellular components, biological processes, and molecular functions. The key genes that are enriched for these GO terms include starch synthesis genes (*OsSSIIa* and *OsSSIIIa*) and cellulose synthesis genes (*CESA2*, *CESA3*, *CESA6*, and *CESA8*). Genes encoding polysaccharides and glutelin were found to be downregulated in the mutant endosperm. The glutelins were further verified by SDS-PAGE, suggesting that glutelin genes could be involved in the null mutant of *OsROS1a* seed phenotype and *OsROS1a* could have the key role in the regulation of glutelins. Furthermore, 378 differentially alternative splicing (AS) genes were identified in the null mutant of *OsROS1a*, suggesting that the *OsROS1a* gene has an impact on AS events. Our findings indicated that the function on rice endosperm development in the null mutant of *OsROS1a* could be influenced through regulating gene expression and AS, which could provide the base to properly understand the molecular mechanism related to the *OsROS1a* gene in the regulation of rice seed development.

## 1. Introduction

The development of seeds is an essential process in the angiosperm life cycle. It involves embryo and endosperm development. The rice endosperm provides energy and materials for seed germination and development, which contains seed storage proteins (SSPs), starch, lipids, and additional trace substances, and occupies most of the space. The production and quality of seed is directly determined by endosperm development at the filling stage. Amino acids, sugars, and other key metabolites’ storage are important for the development of rice endosperm and affect the quality and milling yield [1]. These metabolites are distributed to numerous biosynthetic pathways, mainly the metabolism of starch and the biosynthesis and storage of proteins. In addition, it is also accountable for proteins and starch synthesis in defined amounts and ratios [2].

Rice (*Oryza sativa*) is an excellent material for studying the biosynthesis of SSPs, as it is one of the few plants that synthesize and accumulate both major classes of SSPs, i.e., prolamins and glutelins. The glutelins account for more than 60% of total SSPs in rice, which are encoded by 15 genes, while prolamins are 20% to 30% of total SSPs and are encoded by 34 genes [3,4]. Based on the sequence similarity of the amino acid, glutelins can be divided into four groups (GluA, GluB, GluC, and GluD) [3], and prolamins into three types, 10-, 13-, and 16-kDa [4]. Glutelins are a form of precursor protein (proglutelin), and they synthesize in the endoplasmic reticulum (ER) and transfer the protein body II (PBII) into the protein storage vacuoles (PSV) through the Golgi apparatus [5,6]. Eventually, they are prepared into mature 20-kDa basic and 37-kDa acidic subunits, which are linked by disulfide bonds [7].

The DNA methylation profile study revealed that preferentially expressed genes in the endosperm are mainly coding for major SSPs and starch synthesizing enzymes, and the most important mechanisms for gene activation in rice endosperm are CG and CHG hypomethylation [8]. DNA methylation, an evolutionarily conserved epigenetic mechanism, controls many biological processes such as the imprinting of genes, expression of the tissue-specific genes, stress responses, and transposable elements’ inactivation. The cytosine methylation (5-meC) occurs in three contexts (CG, CHG, and CHH, in which H represents A, T, or C) in plants, and is dynamically regulated by balanced methylation and demethylation [9]. In plants, the active form of DNA demethylation is initialized by the REPRESSOR OF SILENCING (ROS1) transglucosylase gene family, including ROS1, DEMETER (DME), DEMETER-like 2 (DML2), and DML3 [10,11,12]. They all are capable to excise 5-mC, regardless of whether the methylation is in CG, CHG, or CHH form [13,14,15]. Previous studies showed that the loss function of *OsROS1a* produced sterile rice through defects in both female and male gametogenesis [16,17]. *OsROS1a* demethylates both the vegetative cell genome and central cell genome that is vital for viable seed production [18]. The mutation of *OsROS1* generated a new transcript of 21-nt insertion, which increased the number of aleurone cell layers of rice seed by initializing hypermethylation of *rice seed beta-zipper1* (*RISBZ1*) and *rice prolamin*-*box binding factor* (*RPBF*) [19,20]. Furthermore, *ROS1* genes are also involved in the seed development of rice, wheat, and barley by epigenetic influence on the accumulation of SSPs [21,22].

Alternative splicing (AS) is a key regulatory mechanism that directly contributes to the structural and functional diversity of mRNA and proteins [23]. Advancements in high-throughput technology enabled a global analysis of AS, which has been widely discovered in plants, including rice [24], maize (*Zea mays*) [25,26], *Arabidopsis* (*Arabidopsis thaliana*) [27], cotton (*Gossypium raimondii*) [28], pineapple (*Ananas comosus*) [29], and soybean (*Glycine max*) [30]. Furthermore, stage-dependent AS events are probably significantly important, and considerably affect the grain yield significantly. One of the FLOWERING LOCUST (FT) homologues’ gene *FT2* undergoes AS and results in two isoforms in brachypodium (*Brachypodium distachyon*), which work on different functions in the same flowering regulating pathway [31]. In rice, *OsbZIP74* mRNA can be alternatively spliced under treatment with ER stress-inducing agents, which is induced by heat stress, and involved in plant resistance against pathogens or parasites [32]. Based on RNA-Seq, a total of 16,995 AS events of lncRNAs were identified in tomato root, leaf, and flower tissues [33]. More than 1000 genes that experienced AS events were identified in the nitrogen-treated maize roots, and one of the transcription factor *ZmNLP6* isoforms were found to have the strong ability to activate downstream genes [34], suggesting that AS plays a vital role in plant growth and development. There has been emerging evidence showing that DNA methylation can regulate AS. Knockdown of DNA methyltransferase 3 (*Dnmt3*) in honey bees reduced global genomic methylation levels and induced global and diverse changes in AS in fat tissue [35]. The deficiency of DNA methylation in mouse embryonic stem cells has been proven to influence the splicing of more than 20% of alternative exons [36]. However, the effect of the DNA methylation pattern on AS has not been studied in rice.

To better define the role of *OsROS1a* in seed development, the null mutant of *OsROS1a* was generated using the CRISPR/Cas9 system, with 75-nt deletion and 1-nt substitution in the coding region, which destroy the permuted version of a methylated CpG-discriminating CXXC (Per-CXXC) domain, causing alteration in grain size. RNA-Seq analysis revealed that the polysaccharides and glutelin coding genes are downregulated in the 15 days after pollination (DAP) endosperm of the mutant, and 378 genes that experienced AS events were identified, indicating that the function of the rice endosperm development in the null mutant of *OsROS1a* could be influenced through gene expression and AS.

## 2. Results

### 2.1. The Phenotypes of the Null Mutant of OsROS1a 

In order to well describe the role of *OsROS1a* in active DNA demethylation and seed development, we created null mutants of *OsROS1a*, S1 and S2 using the CRISPR/Cas9 system. The homozygous mutant, S1contains an in-frame deletion of 25 amino acid residues from the 1798th amino acid and an amino acid A to T substitution in the Per-CXXC domain, which could lead to the complete loss function of this domain, while the RNA recognition motif fold (RRMF) domain, started from the 1831st amino acid, was maintained (Figure 1). Another biallele gene-editing event was also obtained that has one allele with the same editing as S1 and the other with 86-nt deletion. As the frameshift mutation of *OsROS1a* produces sterile rice, the S2 mutant of T_1_ generation has the same genotype as S1. Therefore, the S1 mutant was used for further analysis.

Microscopic analysis of rice anther showed that T_0_ generation of the null mutant of *OsROS1a* displays partial male sterility with smaller anthers than WT (Figure 2a,g). The S1 mutant anthers had fewer pollen grains as compared to WT, and only few pollen grains were stained by a I_2_-KI solution (Figure 2b,h). To further examine the cellular defects in the S1 mutant, transverse section analysis was performed on the WT and S1 mutant anthers, which were observed at four different developmental stages. At stage 6 (the microspore-mother-cells stage), epidermis, endothecium, tapetum, and the microspore mother cells were clearly visible, and no obvious difference was observed between WT and S1 mutant (Figure 2c,i). However, at stage 9 (the young microspore stage), the microspores in WT were globular, deeply stained, and densely distributed, while the microspores in the S1 mutant were lighter, scarcer, and more irregularly shaped (Figure 2d,j). At stage 10 (the vacuolated pollen stage), as compared to WT microspores that were round and vacuolated, half of the S1 mutant microspores had a normal round shape, comparable to WT, while the remaining microspores were degraded and irregularly shaped (Figure 2e,k). At stage 13 (the mature pollen stage), the WT anther locule was full of mature pollen grains with entirely formed pollen walls and accumulated starch. However, there were fewer microspores in the S1 mutant and some of them appeared to be degenerated and partially sterile (Figure 2f,l). These results are consistent with the pollen staining, indicating that the S1 mutant has partial pollen sterility.

T_1_ generation of the S1 mutant plants were used to examine various agronomic traits. The plant height, panicle length, and the primary branch number were not significantly changed (Figure 3). The plant height was only reduced by 2.4% in the S1 mutant as compared to WT (Figure 3a,c). Similarly, in the case of panicle length, only a 12.8% reduction was observed in the S1 mutant as compared to WT (Figure 3b,e). Likewise, the number of primary branches was reduced by 12.5% (Figure 3f). However, the tiller number was significantly increased (43.9%) (Figure 3d) and the seed fertility percentage was significantly reduced (23.0%) in the null mutant of *OsROS1**a* when compared to WT (Figure 3g).

The null mutant of *OsROS1a* significantly changed the grain size. The seed length was significantly increased by 5.8% and the seed width was significantly decreased by 4.8% in the S1 mutant as compared to WT (Figure 4). To know whether the grain shape and size can affect the seed storage materials, the transverse section analysis was performed for the dehusked mature grains, which involved cutting the seed transversely into two halves. No detectable difference was observed in the aleurone layer between seeds of the S1 mutant and WT. However, the null mutant S1 consisted of deformed seeds containing underdeveloped and less-starch-producing endosperms (Figure 5a), and the shape of embryos was slightly irregular when compared to WT. In contrast to the semitransparent grains of the WT, the S1 mutantgrains were slightly opaque (Figure 5b). The starch granules were further observed by SEM, and the starch granules of WT grains had sharp edges, flat surfaces, and compound and similarly sized polygonal granules. In contrast, the S1 mutant starch granules were variable in size and shape, with rounded and irregular surfaces (Figure 5c). Furthermore, the total starch content was decreased from 72.73% in WT to 69.58% in the S1 mutant, while the amylose content was increased from 11.24% in WT to 12.79% in the null mutant of *OsROS1a* (Figure 6).

### 2.2. RNA-Seq Identifies Responsive Genes in the Null Mutant of OsROS1a

Because the null mutant of *OsROS1a* can affect the rice grain, 15 DAP immature endosperms of the S1 mutant and WT were harvested for RNA-Seq. In total, more than 45,000,000 clean reads were generated for each sample, and the QC30 ratios were all above 94.2%, indicating the transcriptome was of high quality. The abundance of the expressed genes was then quantified using FPKM.

DEGs between WT and the S1 mutant were screened out based on the threshold of the log_2_ fold change being either ≥1 or ≤−1 and a *p*-value < 0.05. Under such criteria, 723 DEGs were identified in total, and among them 290 (40.1%) genes were downregulated and 433 (59.9%) genes were upregulated in the S1 mutant (Figure 7). Furthermore, a GO term enrichment analysis was performed using agriGO [37]. The GO terms with the top 20 enrichment factors were observed, which consisted of eleven biological processes (BP), six molecular functions (MF), and three cellular components (CC). A total of ten out of eleven enriched BP terms are connected to polysaccharide synthetic. In CC, the GO term “macromolecular complex” (GO:0032991) was enriched, including ten glutelin genes (Figure 8).

We found that the expressions of seven starch synthesis- and cellulose-synthesis-related genes were significantly reduced, and most of the starch-synthesis-related genes were downregulated in the S1 mutant (Figure 9, Table 1). Nine of the ten glutelin genes were significantly downregulated, which were then confirmed by SDS-PAGE in dry seeds. The 37-40-kDa acidic and 20-kDa basic two subunits of rice glutelin storage proteins were significantly decreased in the S1 mutant as compared to WT (Figure 10).

### 2.3. RNA-Seq Identifies Differentially AS Events in the Null Mutant of OsROS1a

As DNA methylation can regulate AS, to further investigate whether the 25 amino acid deletion of complete Per-CXXC domain deletion in the demethylase gene *OsROS1a* can regulate AS events occurring in rice 15 DAP endosperm or not, rMATs were used to identify both annotated and novel AS events in the S1 mutant and the expression level of AS events was defined as “exon inclusion level”. In total, 378 differentially (234 up- and 144 downregulated) AS genes were identified, which belonged to SE (Skipped exon), A5SS (Alternative 5’ splice site), A3SS (Alternative 3’ splice site), MXE (Mutually exclusive exon), and IR (Intron retained), which are five major AS types (Figure 11a). Among these, the majority of AS genes occurred via SE, which contains 271 genes (71.69%), and the second most common occurred via IR (58/378, 15.34%) (Figure 11b).

To study the trends in functions of the AS genes, GO enrichment analyses were carried out for genes with differential AS events in rice 15 DAP endosperm. The top 20 enriched GO terms included the largest number of enriched GO terms of 13 in BP, only 1 GO term in CC, and 6 in MF (Figure 11c). In BP, cellular nitrogen compound metabolic process (GO:0034641), nucleobase, nucleoside, nucleotide metabolic process (GO:0055086), and small molecule metabolic process (GO:0044281) are all relevant to small molecule metabolism; cellular localization (GO:0051641), protein localization (GO:0008104), and macromolecule localization (GO:0033036) are all relevant to cellular compound localization. In addition, we found that genes enriched to protein localization (GO:0008104), macromolecule localization (GO:0033036), and establishment of protein localization (GO:0045184) share quite a few common genes, indicating that such a function is likely to be influenced by *OsROS1a* through AS regulation in rice endosperm.

## 3. Discussion

In cereals, the endosperm is a highly specialized storage organ for starch, proteins, and lipids, and plays a crucial role in seed growth and development. Active DNA demethylation in plants is initialized by the ROS1 transglucosylase gene family, including DME, ROS1, DML2, and DML3, which regulates plant developmental processes [10,11,12]. The loss function of *OsROS1a* produced sterile rice through defects in both female and male gametogenesis [16,17]. Furthermore, only two heterozygous lines with one allele of 9-bp and 6-bp produced few seeds, and all frameshift mutants, including six having only truncated RRMF domain, failed to produce seeds [17], suggesting that the RRMF domain is necessary for fertility. We generated the null mutant of *OsROS1a* with 75-bp deletion and 1-bp substitution, which resulted in the complete loss of the Per-CXXC domain but the complete retention of the RRMF domain (Figure 1), which showed semisterile pollen and reduced seed fertility, indicating that the Per-CXXC domain also plays a role in rice fertility. Besides the fertility, the null mutant S1 also altered the rice grain morphology, including long and narrow grain, deformed seeds containing underdeveloped endosperm and a lower amount of starch, slightly irregular shape of the embryo, and slightly opaque grain (Figure 4 and Figure 5). The mutation of *OsROS1a* with a 21-nt insertion generated a new transcript *mOsROS1a* with the insertion of seven amino acid residues (CSNVMRQ) in the RRMF domain to make the thick aleurone and improve rice grain nutrition, which resulted from the hypermethylation and reduced expression of *RISBZ1* and *RPBF* [19,20]. While the expression of *RISBZ1* and *RPBF* was similar between WT and the S1 mutant, this suggested that the Per-CXXC domain cannot alter the expression of these two important TFs. Overexpression of *BiP* suppresses SSPs and starch content and displays the opaque phenotype with shrunken and floury features in rice seeds [38,39].

A total of 723 DEGs were identified in the S1mutant by RNA-Seq analysis. most of these genes are related to starch and SSPs synthesis, suggests that the synthesis of starch, cellulose, and other types of polysaccharide in the endosperm are regulated by *OsROS1a*. The top 20 enrichment factors in GO terms were observed, in which 11 terms consisted of BP, 6 were in MF, and 3 terms were in CC. The BP occupies the largest proportion of GO terms. Out of 11 enriched BP terms, 10 are connected to polysaccharide synthesis. We identified the key genes in starch synthesis, such as *starch synthase IIa* (*OsSSIIa*) and *OsSSIIIa*, and in cellulose synthesis, such as *cellulose synthase A2* (*CESA2*), *CESA3*, *CESA6*, and *CESA8*, that are enriched for these terms, providing more evidence for the hypothesis. *OsSSI* and *OsSSIIIa* contribute to a huge portion of the overall SS enzyme activity in rice during endosperm development [40]. SSIIIa protein linked with other proteins in rice endosperm [41]. Similarly, the *SSIIIa* proportion is also present in a large complex containing *Pyruvate orthophosphate dikinase* (*PPDK*), *ADP-glucose pyrophosphorylase* (*AGPase*), *SSIIa*, and *the starch branching enzyme gene IIa* (*SBEIIa*) and *SBEIIb* in maize [42]. Likewise, *OsSSIIa* enzymes have abundant gene expression in the starch filling stage [43,44], and modify the quality of rice starch [45]. The GO term “macromolecular complex” (GO:0032991) was enriched in the cellular compound and nine glutelin genes related to this term were significantly downregulated in the S1 mutant, which was further confirmed by proteins with SDS-PAGE analysis. The decreased expression of these starch and protein-associated genes in our study suggested that it might be involved in the null mutant of *OsROS1a* seed phenotype and *OsROS1a* might have a main role in the regulation of these genes. The study of the DNA methylation profile during the development of endosperm revealed that genes expressed preferentially in endosperm were related to key storage proteins and starch synthesizing enzymes, are normally hypomethylated [8], and suggested that the demethylation of CG and CHG was the main mechanism for gene activation.

AS is an essential post-transcriptional process that generates several mRNA variants from a single pre-mRNA molecule and improves the genome coding and regulatory potential. Several studies have been carried out to find the AS events in different plant species, tissues, and environmental conditions [32,33,34]. However, the effect of the DNA methylation pattern on AS has not been reported in rice. Advancements in high-throughput technology enabled a global analysis of AS to study its functional characteristics in response to stress [46]. In this study, we identified 378 differentially AS genes in the 15 DAP endosperm of the S1 mutant, which belonged to SE, A5SS, A3SS, MXE, and IR, the five major AS types (Figure 11a), and the SE (71.69%) and IR (15.34%) were the top two types (Figure 11b). Whereas in *Arabidopsis*, AS had the low level of SE (<5%) and the high level of IR (65%) [47]. The SE and IR were predominant AS events in maize endosperm containing 28.33% and 29.85%, respectively [48]. Earlier studies have revealed that AS changes are affected by developmental and environmental factors [49,50]. The SE type is comparatively low (5%) in most maize tissues, which can be increased by up to more than 27% under abiotic stress. Furthermore, during seed development of maize, AS isoforms alter considerably in seed, embryo, and endosperm [25]. Our results revealed that AS genes of the A5SS and the MXE types were relatively less common and less influenced by the mutation of *OsROS1a* during endosperm development and that the function modification of *OsROS1a* did not lead to AS type-specific. The mutation of the *OsROS1a* gene showed a major impact on the alternation of AS events of downregulated genes and the impacts are not AS type-specific in the 15 DAP rice endosperm. Several GO terms indicated identical functions were enriched. Moreover, we identified that *OsROS1a* might influence the genes which are enriched to “protein localization” (GO:0008104), “macromolecule localization” (GO:0033036), and “establishment of protein localization” (GO:0045184) through AS regulation in the endosperm. These results suggested that rice endosperm development mediated by *OsROS1a* gene could be influenced through AS regulating.

## 4. Materials and Methods

### 4.1. Generation of Null Mutants of ROS1a in Rice Using CRISPR/Cas9 System

The CRISPR/Cas9 binary vector, pYLCRISPR/Cas9 Pubi-H, was used for targeted genome editing [51]. Two single guide RNA (sgRNA) were designed to precisely target the 13th exon of *OsROS1a*. Two sgRNA intermediate vectors, pYLsgRNA-OsU6a and pYLsgRNA-OsU6b, were used for gene targeting on rice OsU6a and OsU6b small nuclear RNA promoters, respectively. The sgRNA sequences were cloned into a binary vector that contained sgRNA and Cas9 expression cassettes, and the resulting construct was transformed into rice cultivar Nipponbare by *Agrobacterium* infection of callus explants.

### 4.2. Pollen Fertility Examination and Histochemical Assay

Floral organs were photographed with a dissecting microscope. For the analysis of pollen fertility, the anther from the WT (wild type, Nipponbare) and *osros1a* mutant plants were sampled from the spikelets just before flowering and the 1% potassium iodide (I_2_-KI) solution was used to stain the pollen grains. A Nikon eclipse Ni fluorescence microscope was used to be visualized, and the stained pollen grains was photographed. For transverse section analysis, spikelets of various anther development stages from WT and the S1 mutant, according to the previous study [52], were collected and frozen in an OCT (Tissue-Tek; Sakura Finetek, Torrance, CA, USA) compound. Samples were sectioned with the help of the freezing microtome (Thermo Shandon Cryotome FE, Shandon, China). Then, 0.05% toluidine blue dye was used to stain the crossed section and the microscopic images were captured by a Nikon eclipse Ni fluorescence microscope.

### 4.3. Agronomic Traits Analyses

The agronomic traits of plant height, panicle length, tiller number, fertile seed percentage, and the primary and secondary branches’ number in the main panicle of the *osros1a* mutant and WT were measured for plant grown in the greenhouse.

### 4.4. Half-Seed Assay and Scanning Electron Microscopy (SEM)

For Evans Blue staining-based half-seed assay, mature grains from WT and the null mutant S1 were dehusked and transversally sectioned into two halves using a razor blade, then dipped in 0.1% (*w*/*v*) Evans Blue solution for 10 min, and by using distilled water, were washed three times. All samples were observed and photographed under a dissecting microscope (Nikon). Further, to study the mature seeds’ morphological changes and their starch granules, the WT and the S1 mutant seeds were transversely sectioned. Samples were then examined and photographed by SEM.

### 4.5. RNA Extraction, Library Preparation, and RNA-Seq

The S1 mutant was cultivated in the greenhouse, and the extraction of total RNA from the mutant and WT endosperm was conducted at 15 DAP. For RNA-Seq analysis, 1 μg RNA per sample was used. The mRNA was enriched using magnetic beads with oligo-dT, then divalent cations in NEBNext First Strand Synthesis Reaction Buffer (5X) were used to break it into short fragments and then reversed transcription into cDNA. DNA fragments 3′-ends were adenylated, and a hairpin loop structure NEBNext Adaptor was ligated to prepare for hybridization. After purification, terminal modification, fragments selection, and the amplification of PCR with Phusion High-Fidelity DNA polymerase and another round of purification, two libraries (one for WT and one for the mutant) were constructed for Next-Generation Sequencing (NGS). The Agilent Bioanalyzer 2100 system was used to assess the library quality.

A cBot Cluster Generation System was utilized for performing clustering of the index-coded samples, using TruSeq PE Cluster Kit v3-cBot-HS (Illumia) as per the manufacturer’s instructions. After cluster generation, the preparations of the library were sequenced on an Illumina Novaseq platform and 150 bp paired-end reads were generated.

### 4.6. Quality Control and RNA-Seq Analysis

The clean reads were obtained by removing reads containing ploy-N, adapter, and low-quality from raw data. Hisat2 (v2.0.5, Daehwan Kim, Dallas, TX, USA) was adopted to build the reference genome index and align the pair-end clean read to the genome [53]. We chose IRGSP1.0 (https://rapdb.dna.affrc.go.jp/download/irgsp1.html, accessed on 20 December 2020), as the reference genome for its high quality and comprehensive annotation. Samtools (v1.4.1, Heng Li, London, UK) was then used to convert the SAM (Sequence Align Mapping) file to its binary coding form BAM file [54], which was smaller and faster to perform downstream analysis on.

To count the reads numbers mapped for each gene, Feature Counts (v1.5.0-p3, Yang Liao, Melbourne, Australia) was used [55]. The FPKM (Fragments Per Kilobase of transcript sequence per Millions base pairs sequenced) of each gene was then calculated based on the gene length and reads count mapped to this gene. DESeq2 R package v1.26.0 was then used to determine DEGs (Differential Expression Genes) [56]. The threshold for significant differential expression was set as fold change ≥2 and *p*-value ≤ 0.05. GO (Gene Ontology) enrichment analysis of DEGs was implemented using agriGO (v2.0, Tian Tian, Beijing, China) [37].

### 4.7. SDS-PAGE Analysis

Storage protein analysis was conducted by SDS-PAGE with the Laemmli method. Mature rice seeds of the S1 mutant and WT were ground with the help of mortar and pestle to make it fine powder. Rice glutelin was extracted from powdered seeds (25 mg) by 0.2% NaOH after stepwise removal of albumins (with deionized water), globulins (with 0.5 M NaCl and 50 mM Tris-HCl; PH 6.8), and prolamins (with 70% (*v*/*v*) alcohol). SDS-PAGE of 12% gel was conducted by using Standard Twin Mini Gel Unit (CA, USA) for 2.5 h at 120 V. The glutelin proteins were detected by staining the gel with a staining solution having Coomassie Brilliant Blue R-250 (Sigma) for 30 min and then 20% methanol, and 5% acetic acid in DW solution was used for destaining. The protein bands of the sample were measured by comparing it with the protein ladder (Thermo Scientific Page Ruler Prestained Protein Ladder 10-250 KDa) in the electropherogram.

### 4.8. Identification of Differential AS Events

To investigate the difference in AS pattern between WT and mutant, rMATs (v4.1.1, Shihao Shen, CA, USA) was used to identify both annotated and novel AS events in the mutant [57], and the expression level of AS events was defined as “exon inclusion level”. The threshold was set to 0.0001 and AS events with a *p*-value above 0.05 were filtered out for further analysis. As a result, five major types of AS events, including SE, A5SS, A3SS, MXE, and IR, were identified. GO enrichment analysis of DAGs (Differential Alternative splicing Genes) was implemented using DAVID [58].

## 5. Conclusions

In summary, we identified that the loss function of the *OsROS1a* gene altered the rice grain size, having longer grain and a reduced width as compared to WT. The null mutant of *OsROS1a* seeds were deformed containing underdeveloped, less-starch-producing endosperm and a slightly irregularly shaped embryo. Furthermore, RNA-Seq analysis showed that many genes encoding polysaccharides and glutelins were found to be downregulated in the endosperm of the S1 mutant, suggesting that it might be involved in the S1 mutant seed phenotype and *OsROS1a* might have a main role in the regulation of these genes. Moreover, AS analysis revealed that the *OsROS1a* gene has a major impact on the alternation of AS events of these downregulated genes. These findings have provided a base to properly understand the molecular mechanism related to the *OsROS1a* gene in the regulation of rice seed development.

## Figures and Tables

**Figure 1 ijms-23-06357-f001:**
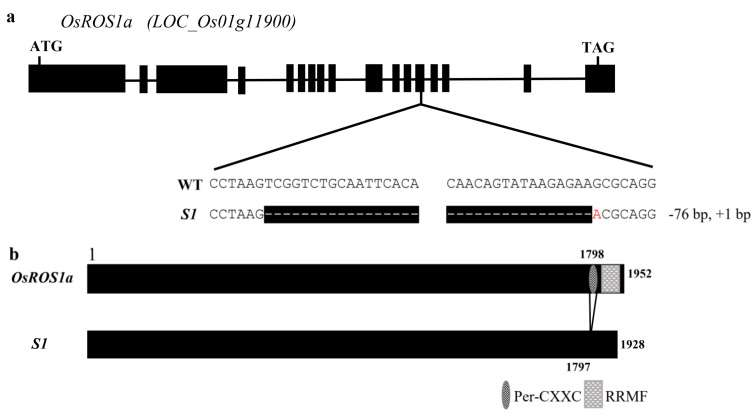
CRISPR/Cas9-induced *OsROS1a* gene editing. (**a**) Schematic of *OsROS1a* gene structure. Exons and introns are denoted as black blocks and lines, respectively. The translation initiation codon (ATG) and the termination codon (TAG) are shown. The recovered mutated allele is shown below the WT reference sequences. The target sites’ nucleotides are indicated in black capital letters. The white dashes indicate the deleted nucleotides, and the substitution nucleotide A is shown in red. (**b**) The protein structure of WT and the null mutant S1. The predicted structure of WT contained the Per-CXXC domain and RRMF domain, while the Per-CXXC domain is not predicted inthe null mutant S1.

**Figure 2 ijms-23-06357-f002:**
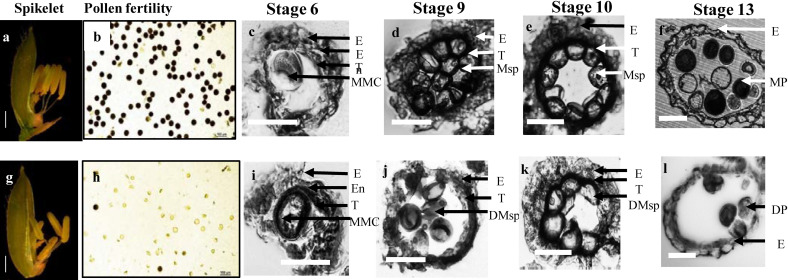
The phenotype of S1 mutant. The spikelet of WT (**a**) and S1 mutant (**g**). The pollen fertility of WT (**b**) and the S1 mutant (**h**) by iodine staining magnified 20×. Transverse section analysis of the anther development in WT (**c**–**f**) and the S1 mutant (**i**–**l**). Locules from the anther section of WT and at stage 6 (the microspore mother cells stage), stage 9 (the young microspore stage), Stage 10 (the vacuolated pollen stage), and stage 13 (the mature pollen stage). E, epidermis; En, endothecium; T, tapetum; MMC, microspore mother cells; Msp, microspores; MP, mature pollen; DMsp, degenerated microspores; DP, degenerated pollen. Scale bar: 50 μm.

**Figure 3 ijms-23-06357-f003:**
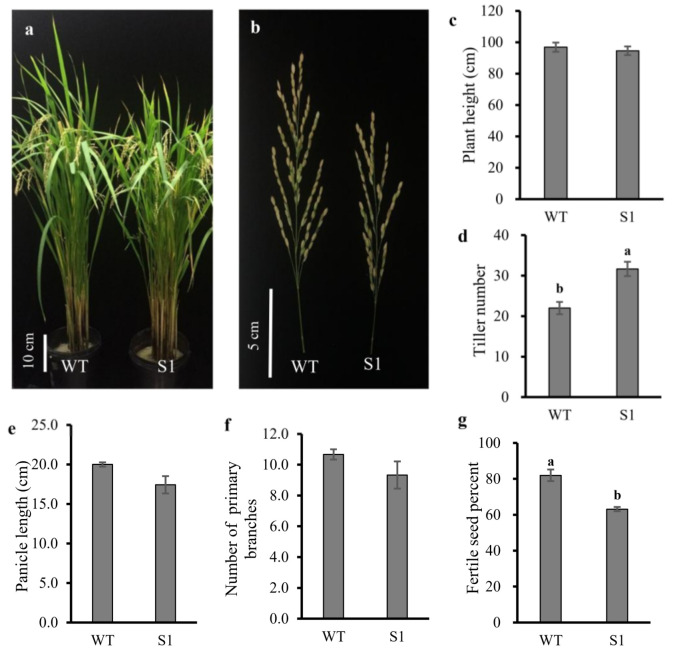
Phenotypes of WT and the S1 mutant. (**a**) Phenotype comparison of the WT and the S1 mutant(**b**) The panicle of WT and the S1 mutant. The statistic calculation of agronomic traits of (**c**) plant height, (**d**) tiller number, (**e**) panicle length, (**f**) number of primary branches, and (**g**) fertile seed percentage. Lettering indicates statistical significance at *p* ≤ 0.01. Data are means ± standard deviation (SD) (*n* = 3).

**Figure 4 ijms-23-06357-f004:**
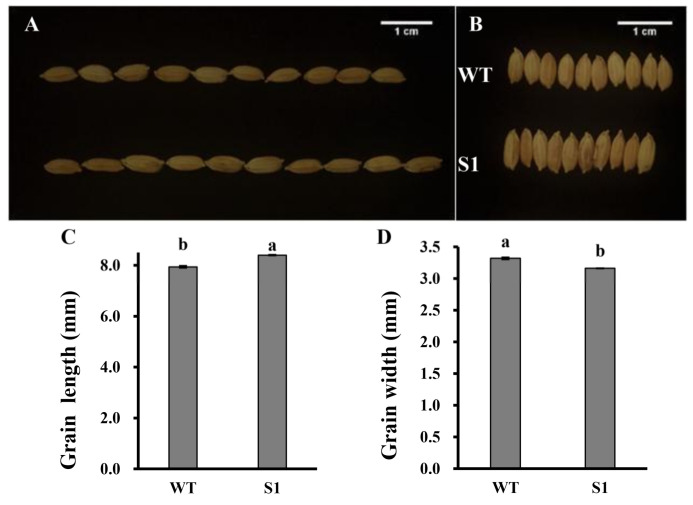
Comparison of seed length (**A**) and seed width (**B**), and the statistical analysis of seed length (**C**) and seed width (**D**) between WT and the S1 mutant. Lettering indicates statistical significance at *p* ≤ 0.01. Data are means ± SD (*n* = 3).

**Figure 5 ijms-23-06357-f005:**
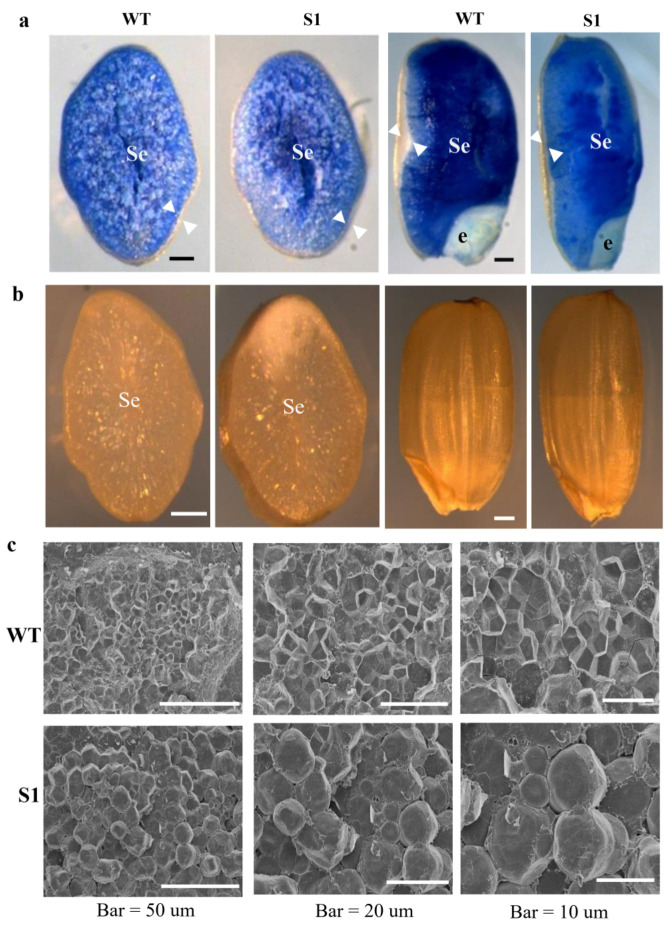
Genetic screening of *OsROS1a* gene editing mutant by half-seed assay. (**a**) Transversally and longitudinally sectioned of WT and the S1-mutant dehusked, mature grains, stained with the Evans blue dye. Arrowheads indicate the aleurone. se, starchy endosperm; e, embryo. (**b**) Transversely sectioned and nonsectioned mature grains showed the opaque endosperm in the S1 mutant, as compared to the semitransparent endosperm in WT. Scale bar, 0.5 mm. (**c**) Scanning electron micrographs of the endosperm in transverse sections are shown with increasing magnification from left to right. Scanning electron microscopy (SEM) of WT grain revealed similarly sized polygonal starch granules with sharp edges; smooth, flat surfaces; and compound starch granules while the S1 granules are rounded, variable in size and shape, and have uneven surfaces.

**Figure 6 ijms-23-06357-f006:**
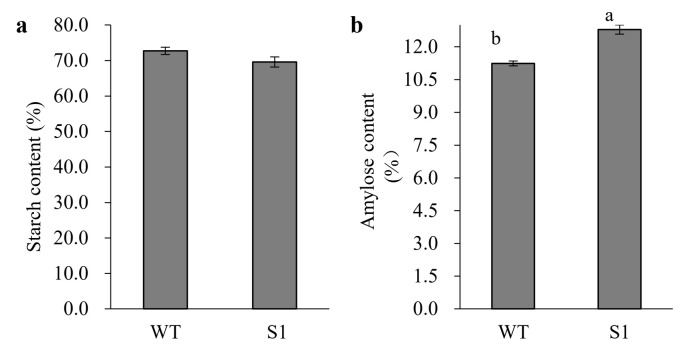
Total starch and amylose content in the mature seed of WT and the S1 mutant. (**a**) Total starch content in the seeds of WT and S1 mutant. (**b**) Total amylose content in seeds of WT and S1 mutant. Lettering indicates the statistical difference at *p* ≤ 0.01. Data are means ± SD with three replicates.

**Figure 7 ijms-23-06357-f007:**
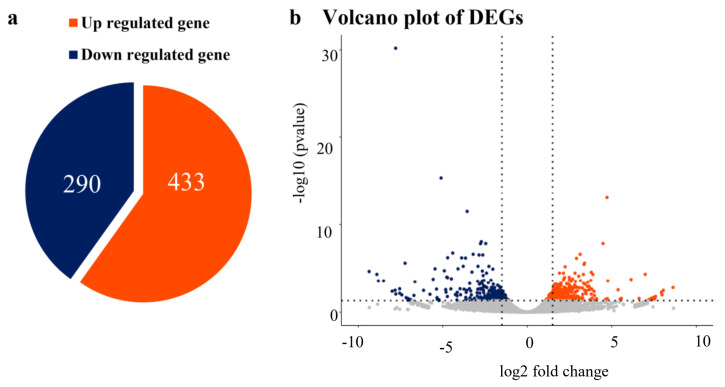
Differential expression of 15 DAP endosperm from WT and the S1 mutant. (**a**) The number of upregulated genes and downregulated genes. (**b**) Volcano plot of differentially expressed genes with the threshold of |log_2_ foldchange| > 1. Orange indicates upregulated genes, and blue indicates downregulated genes.

**Figure 8 ijms-23-06357-f008:**
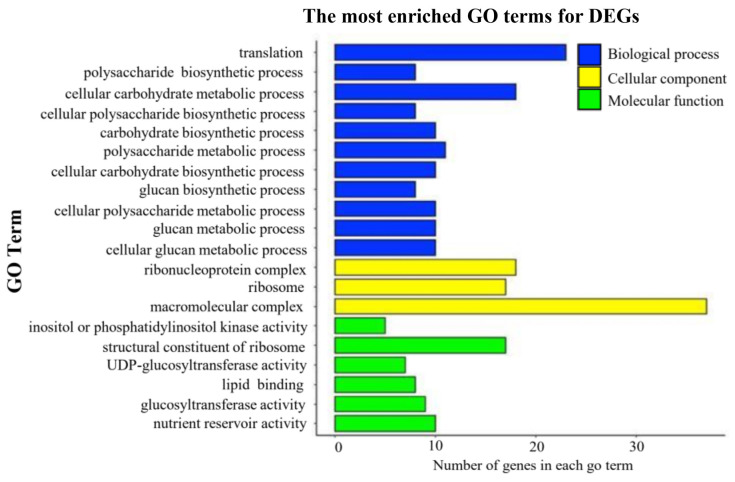
GO enrichment of DEGs. GO-term enrichment analysis was performed using agriGO [37], and the top 20 GO terms were shown that belonged to biological processes, cellular components, and molecular functions.

**Figure 9 ijms-23-06357-f009:**
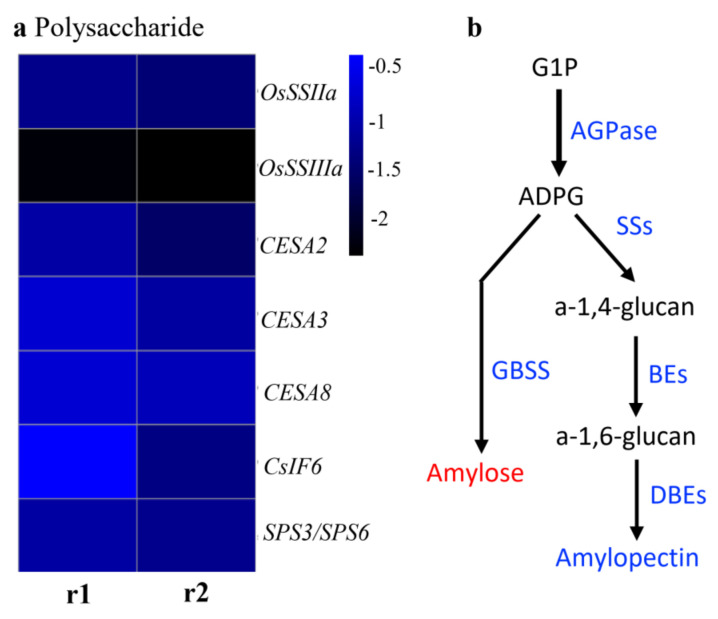
The expression of polysaccharide-related genes and starch synthesis pathway. (**a**) The heat plot of differential expressed polysaccharide-related genes. (**b**) The starch synthesis pathway. The downregulated genes were shown in blue colors, amylose in red showed increased content, and amylopectin in blue showed decreased content.

**Figure 10 ijms-23-06357-f010:**
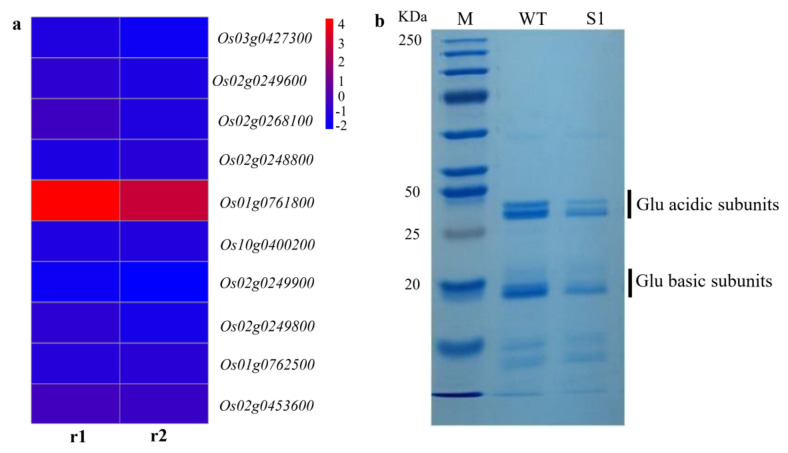
The gene expression and protein accumulation of glutelins. (**a**) The heat plot of significantly differentially expressed glutelin genes based on the RNA-Seq data. (**b**) SDS-PAGE analysis of glutelins. Glutelins extracted from mature seeds of WT and the S1 mutant were separated on a 12% SDS-PAGE gel and stained with Coomassie Brilliant Blue (CBB).

**Figure 11 ijms-23-06357-f011:**
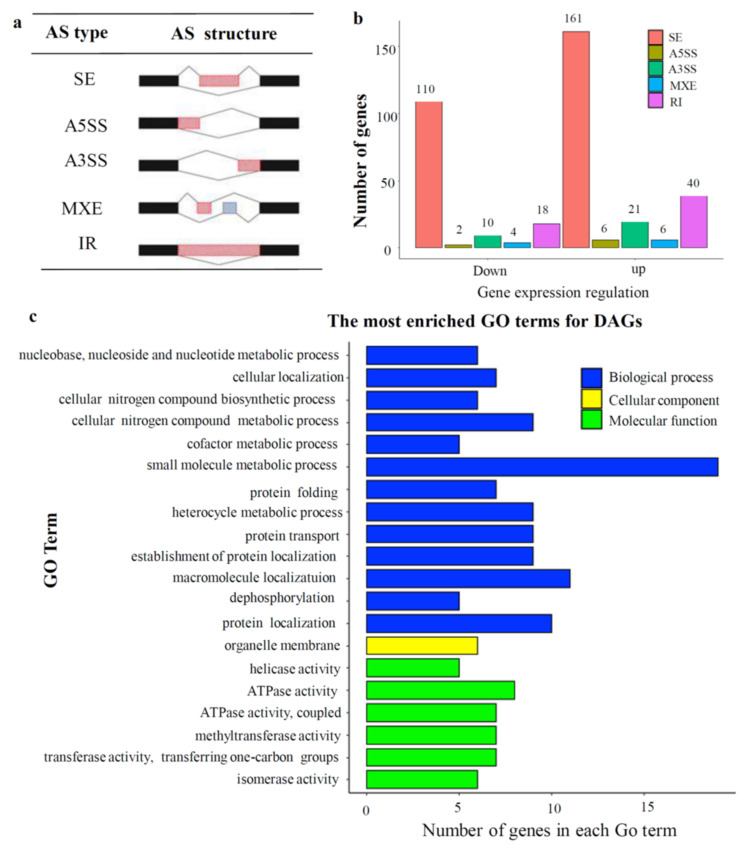
Alternative splicing affected by the mutation of *OsROS1a*. (**a**) Five types of alternative splicing events. (**b**) Statistics of differential alternative spliced genes between WT and the S1 null mutant. (**c**) GO enrichment of differential alternative spliced genes.

**Table 1 ijms-23-06357-t001:** Expression values of starch and cellulose-synthesis-related genes.

Gene Name	S1-r1	S1-r2
*OsSSIIa*	−1.358472011	−1.538954326
*OsSSIIIa*	−2.395417526	−2.467690768
*CESA2*	−1.163670981	−1.655416381
*CESA3*	−0.825392927	−1.18339011
*CESA8*	−0.792395169	−1.006805408
*Cs1F6*	−0.450447312	−1.428153108
*sps3*/*sps6*	−1.168312567	−1.35323233

## Data Availability

The data presented in this study are available within the article.

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
