# Peer review of "The Function of DNA Demethylase Gene ROS1a Null Mutant on Seed Development in Rice (Oryza Sativa) Using the CRISPR/CAS9 System"

_ijms, 2022, doi:10.3390/ijms23126357_

Round 1

Reviewer 1 Report

The manuscript titled "The Function of DNA Demethylase Gene ROS1a Null Mutant 2 on Seed Development in Rice (Oryza Sativa) Using the 3 CRISPR/CAS9 System" created a nll mutant of DNA demethylase gene, the OsROS1a, by CRISPR/Cas9 technology in rice. They found that the edosperm development was inhibited in grains of ros1a mutant, including the glutelin and starch biosynthesis. RNA-seq revealed that gene expressions involved in starch bisynthesis and glutelin formation were downregulated in ros1a mutant during grain filling. Their results demonstrated that OsROS1a gene plays key role in regulating seed development. The topic of this manuscript is very interesting and the experiments were well designed. The results were well analyzed and presented. The manusctipt was well written and is acceptable in the current version. Here is one minor suggestion: the authors have shown the RNA-seq results of some key gene invovled in starch and glutelin biosynthesis, but no qRT-PCR validating results were shown in this manuscript. Besides, the enzyme activites of starch synthase or other enzymes related to starch biosynthesis can be assayed to further support their conclusion.

Author Response

Thanks very much for the reviewer’s comments. It is a very good suggestion. The expression of key genes and the enzyme activities will be further investigated, and we will go forward to validate the functions of these key genes using CRISPR/cas9 gene editing system. We think it could be enough of RNA-Seq data that showed differentially expressed in this manuscript.

Reviewer 2 Report

The manuscript by Irshad et al entitled “The Function of DNA Demethylase Gene ROS1a Null Mutant on Seed Development in Rice (Oryza Sativa) Using the CRISPR/CAS9 System” has highlighted the regulatory role of OsROS1a in regulation of rice seed development. They also indicated the role of OsROS1a alternative splicing events. Overall the manuscript is fine, the experiments are conducted well and the results as nicely corelated. I do not have any major concern with the present format of manuscript, however, I have few suggestions which are mentioned below:

-          Authors needs to elaborate the abbreviation at the first place it is used (eg. 63, 70, 77, and many others).

-          What I observed is that, the pollen phenotype is stronger than that of the seed, thus I would like to suggest authors to go for quantification of pollen viability and then they can corelate the phenotype.

-          As, authors have used a single CRISPR line, authors should discuss how they have eliminated the possibility that the observed phenotype is mainly associated with the ROS1a and not to off-target effect. This would be helpful for the readers.

-          Please go through the entire manuscript for linguistic and grammatical errors (Eg. 246)

-          Discussion need to be improved, as it should be more focused on the observed phenotype.

-          Authors can incorporate a conclusion section as well, as it will be in the benefits of the readers.

Author Response

  1. Authors needs to elaborate the abbreviation at the first place it is used (eg. 63, 70, 77, and many others).

Response: Thank you very much for pointing out the problem. We carefully went through the whole manuscript and elaborated the abbreviation as the reviewer suggested.

  1. What I observed is that, the pollen phenotype is stronger than that of the seed, thus I would like to suggest authors to go for quantification of pollen viability and then they can corelate the phenotype.

Response: Very good suggestion. When we got the pollen of T1 generation, it is not strong. We also obtained the frameshift knockout osros1a mutants that are complete sterile. We will have another manuscript to focus on the pollen phenotype.

  1. As, authors have used a single CRISPR line, authors should discuss how they have eliminated the possibility that the observed phenotype is mainly associated with the ROS1a and not to off-target effect. This would be helpful for the readers.

Response: Thanks for your comment. We got another bi-allele event with 75-nt and 86-nt deletion, both events had the same phenotypes and we think it shouldn’t be off-target effect, which we added in the revised version.

  1. Please go through the entire manuscript for linguistic and grammatical errors (Eg. 246)

Response: Thanks very much for your constructive suggestion. We carefully went through the whole manuscript and revised the linguistic and grammatical errors to make it to be suitable for publication in this journal.

  1. Discussion need to be improved, as it should be more focused on the observed phenotype.

Response: Very good suggestion. We have revised the Discussion section as the reviewer suggested and deleted the observed phenotype parts.

  1. Authors can incorporate a conclusion section as well, as it will be in the benefits of the readers.

Response: Thanks very much for the reviewer’s comment. The conclusion has been edited as the reviewer suggested.